# Artificial Intelligence Algorithm to Screen for Diabetic Neuropathy: A Pilot Study

**DOI:** 10.3390/biomedicines13051075

**Published:** 2025-04-29

**Authors:** Giovanni Sartore, Eugenio Ragazzi, Francesco Pegoraro, Mario German Pagno, Annunziata Lapolla, Francesco Piarulli

**Affiliations:** 1Department of Medicine-DIMED, University of Padova, 35122 Padova, Italy; g.sartore@unipd.it (G.S.); f.pegoraro@icloud.com (F.P.); mariogerman.pagno@studenti.unipd.it (M.G.P.); annunziata.lapolla@unipd.it (A.L.); francesco.piarulli@unipd.it (F.P.); 2Studium Patavinum, University of Padova, 35122 Padova, Italy

**Keywords:** type 2 diabetes, diabetic polyneuropathy, biothesiometry, artificial intelligence, risk evaluation, Cohen’s kappa

## Abstract

**Background/Objectives**: Patients with type 2 diabetes (T2D) are at risk of developing multiple complications, and diabetic polyneuropathy (DPN) is by far the most common. The purpose of the present study was to assess the ability of a new algorithm based on artificial intelligence (AI) to identify patients with T2D who are at risk of DPN in order to move on to further instrumental evaluation with the biothesiometer method. **Methods**: This is a single-centre, cross-sectional study with 201 consecutive T2D patients recruited at the Diabetes Operating Unit of the ULSS 6 of Padua (Northeast Italy). The individual risk of developing DPN was calculated using the AI-based MetaClinic Prediction Algorithm and compared with the DPN diagnosis provided by the digital biothesiometer method, which measures the vibratory perception threshold (VPT) on both feet. **Results**: Of the enrolled patients, 107 (53.23%) were classified by AI software as having a low probability of developing DPN, 39 (19.40%) as having a moderate probability, 29 (14.43%) as having a high probability, and 26 (12.94%) as having a very high probability. In 63 of the total patients, biothesiometer measurement showed a VPT ≥ 25 V, indicative of DPN, while 138 patients had a non-pathological VPT value (< 25 V) (prevalence of abnormal VPT 31.34%; prevalence of normal VPT 68.66%). The overall agreement between biothesiometer results and AI risk attribution was 65%. Cohen’s κ was 0.162, and Gwet’s AC1 coefficient 0.405. **Conclusions**: The use of an optimized AI algorithm can help estimate the risk of developing DPN, thereby guiding more targeted and in-depth screening, including instrumental assessment using the biothesiometer method.

## 1. Introduction

Patients with type 2 diabetes (T2D) are often at risk of developing multiple comorbidities [1], such as stroke, heart attack, blindness, and amputation [2], lowering life expectancy. The global diabetes epidemic has led to a corresponding exponential increase in its complications, and among these is neuropathy, of which distal symmetric polyneuropathy (diabetic polyneuropathy, DPN) is by far the most common. This complication, particularly when it leads to foot ulcers or painful neuropathy, has a profound impact on the patient’s quality of life [3,4]. DPN may be associated with other diffuse neuropathies secondary to diabetes, which include a variety of autonomic neuropathies (such as cardiac autonomic neuropathy, gastrointestinal dysmotility, or diabetic cystopathy) [5]. During their lifetime, at least 50% of individuals with diabetes mellitus develop DPN [5].

The incidence of DPN is higher in people with T2D (6100 per 100,000 person-years) than in those with type 1 diabetes (T1D) (2800 per 100,000 person-years) [6,7]. In contrast, neuropathy prevalence appears comparable between patients with T1D (11–50%) [8,9,10] and those with T2D (8–51%) [9,11,12]. The prevalence is even higher if asymptomatic diabetic neuropathy is included. In this case, 45% of patients with T2D and 54% of those with T1D would be affected [9]. The prevalence of diabetic neuropathy also varies with the duration of the disease. For example, during a 10-year follow-up, the prevalence of DPN increased from 8% to 42% in patients with T2D [10]. Without radical intervention, it is estimated that by 2050, out of the projected 9.7 billion people worldwide, 1/6 will suffer from DPN [13].

The American Diabetes Association and the Canadian Diabetes Association currently recommend screening for DPN at diagnosis for T2D patients and five years after diagnosis for T1D patients; thereafter, screening is annual for all patients [7,14,15]. Screening tests must be rapid, reliable, simple, and reproducible. In DPN, a very early clinical sign is the loss of vibratory sensitivity (carried by larger nerve fibres). One of the most widely used tools to evaluate quantitative data is the biothesiometer, consisting of a probe applied to the skin that generates increasing intensities of vibration [16,17].

Given the complex and multifactorial nature of DPN, a promising strategy for the early diagnosis of diabetic complications involves the use of artificial intelligence (AI) algorithms to identify the patients most at risk of complications [18]. Subjects most at risk can be enrolled in more stringent screening, therefore avoiding or at least delaying the onset of complications [19,20,21]. Nowadays, the use of AI models or algorithms is still at an embryonic stage. Such tools are not routinely used in daily clinical practice [22]. Few studies in the literature have explored the role of AI in the diagnosis and management of diabetes mellitus complications [23]. Even fewer studies have investigated the use of AI for the screening of DPN or diabetic foot ulcers [24]. Deep learning algorithms have been successfully applied to classify DPN using imaging data such as corneal confocal microscopy, demonstrating high performance in both development and validation cohorts [25,26,27]. Beyond imaging, AI models have been integrated with clinical neurophysiological data and metabolic parameters to diagnose and predict the future occurrence of polyneuropathy in individuals with diabetes and prediabetes, offering a non-invasive and scalable approach to early intervention [28]. Moreover, AI combined with neuroimaging techniques has shown promise in assessing treatment response in painful DPN, pointing to its potential in personalized care [29]. Additional studies have explored clinical decision-support systems and automated diagnostic tools based on deep learning, further underscoring the versatility of AI in addressing the complexities of DPN diagnosis and prognosis [30,31].

The purpose of the present study was to assess the ability of a new AI-based algorithm [23] in identifying patients with T2D who are at risk of DPN in order to proceed with further instrumental evaluation using the biothesiometer method. The AI tool used in our study is designed primarily for early risk stratification, with the goal of identifying individuals at increased risk of developing DPN even before clinical symptoms become evident, with an estimated risk horizon of approximately 12 to 24 months. The AI model is intended not only to aid in the diagnosis of established DPN but, more importantly, to support proactive care by identifying patients at risk prior to symptom onset, thus contributing to improved clinical outcomes and healthcare efficiency.

## 2. Materials and Methods

### 2.1. Patients

This is a single-centre, cross-sectional study in which consecutive patients were recruited at the Diabetes Operating Unit of the ULSS 6 of Padua (Northeast Italy). Individuals who met the following inclusion criteria at the visit were considered eligible: age between 18 and 80 years, diagnosis of T2D, absence of symptoms or diagnosis of peripheral neuropathy, Caucasian ethnicity, availability of clinical and biochemical data to calculate the probability of onset of DPN. Subjects who met the following criteria were excluded from the study: central or peripheral neurological diseases of any type, diagnosis or symptoms of peripheral neuropathy, alcohol abuse, chronic therapy with neuroleptics or other neurotropic drugs, pregnancy, or breastfeeding. To align with the intended use of the AI prediction tool as a screening instrument for asymptomatic patients, we excluded individuals with any self-reported or clinically documented symptoms of peripheral neuropathy. This design choice was made to reduce diagnostic bias and to ensure a more objective evaluation of the tool’s ability to flag risk prior to overt clinical manifestation. The goal was not to confirm neuropathy in already-symptomatic patients, but rather to assess the model’s performance in identifying those who may benefit from early diagnostic follow-up. The study was conducted in accordance with the Declaration of Helsinki and its later amendments and was approved by the local ethics committee of the province of Padova (study No. 3884/U16/16, protocol No. 0045112, approval date 14 July 2016). The study was conducted in accordance with applicable data protection regulations, including the principles outlined in the General Data Protection Regulation (GDPR). The data used consisted exclusively of already available clinical records (anthropometric and biochemical parameters), along with the biothesiometer non-invasive measurement of peripheral nerve sensitivity performed at a single time point. No additional procedures or data collection beyond standard care were involved. All data were anonymized prior to analysis, and no identifiable personal information was used in the AI evaluation. This cross-sectional, observational study design ensured minimal risk to participants, and appropriate safeguards were implemented to maintain data security and confidentiality throughout. Informed consent was obtained from all subjects involved in the study.

### 2.2. Anthropometric and Biochemical Parameters

At the medical visit, the following data were collected: age, sex, disease duration, body mass index, systolic and diastolic blood pressure, glycated haemoglobin (HbA1c), fasting blood glucose, total cholesterol, HDL cholesterol, LDL cholesterol, triglycerides, aspartate aminotransferase (AST), alanine aminotransferase (ALT), creatininemia, albuminuria. The biochemical parameters were measured, according to standard laboratory procedures, in the morning after an overnight fast. Albuminuria was measured in morning spot urine.

### 2.3. Artificial Intelligence (AI) Software

MetaClinic (METEDA S.r.l., Rome, Italy) is an artificial intelligence software that calculates a probability of organ damage onset within 2 years relative to the six main target organs (heart, peripheral vessels, cerebral vessels, eyes, kidney, peripheral nerves) involved in diabetes complications. The tool is implemented in the electronic medical record system available at diabetes care centres in Italy and aims to support the physician in the early identification of subjects potentially at risk of complications. The AI software for predicting diabetes complications was originally described by Nicolucci and colleagues [23], who used XGBoost-based models [32] that were trained on the Smart Digital Clinic EMR data (Smart Digital Clinic, METEDA S.r.l., Rome, Italy) from 147,664 patients across 23 Italian diabetes centres over 15 years. To ensure true independence, models were externally validated on five additional different diabetes centres whose data were never part of the training set, thereby preserving generalizability and preventing data leakage. The sample size of the validation cohorts ranged from 3912 to 20,007 patients. Our study considered a totally independent population. The algorithm operates for patients aged ≥ 18 years with a diagnosis of T2D using at least 40% of a predefined pool of 25 clinical or biochemical parameters available in the last year: height, weight, BMI, waist circumference, diastolic pressure, systolic pressure, fasting blood glucose, post-breakfast blood glucose, post-lunch blood glucose, platelets, haemoglobin, glycated haemoglobin (HbA1c), creatinine, creatinine clearance, microalbuminuria, creatine phosphokinase (CPK), LDL cholesterol, estimated LDL cholesterol, HDL cholesterol, total cholesterol, triglycerides, aspartate aminotransferase (AST), alanine aminotransferase (ALT), gamma-glutamyl transferase (GGT), and uric acid. The AI prediction of diabetes-related complication is presented according to four levels: low (risk < 10%), moderate (risk 10% to 25%), high (risk > 25% to 50%), very high (risk > 50%) [33]. These categories are defined by applying 3 percentile cutoffs (the 50th, 75th, and 90th percentiles) to the calibrated risk probabilities in the short term (2 years) and medium term (3–5 years) for each complication, as described in the training and validation workflow [23]. The AI prediction in this study was focused on somatosensory peripheral diabetic neuropathy; since the comparison was performed with instrumental biothesiometer measurement, the autonomic diabetic neuropathy was not considered.

### 2.4. Peripheral Neuropathy Measurement

Once patients were enrolled, the individual risk of developing DPN was calculated using the tool integrated into the clinical program MetaClinic Prediction Algorithm for Windows (version 10.13.8.0 or later, METEDA S.r.l., Rome, Italy). Subsequently, during the same medical examination, the presence or absence of DPN was confirmed by measuring the vibratory perception threshold (VPT) on both feet using a digital biothesiometer (METEDA S.r.l., Rome, Italy). This is a portable, hand-held medical device with a plastic tip that vibrates at 100 Hertz. Vibration is electronically calibrated to ensure constant amplitude, avoiding the variable represented by tissue density at the measurement site. Vibration amplitude (1 to 10.5 microns, proportional to the voltage, 5–32 V) may be set by the operator or automatically regulated in order to assess the patient’s sensitivity. After familiarizing the patient with the sensation of vibration by holding the instrument turned on with the tip on the surface of the palm of the hand, the tip was then placed perpendicularly to the metatarsophalangeal joint of the first toe. The vibration amplitude was slowly and steadily increased (1 mV/s), and the VPT value was defined as the value at which the patient reported perceiving the vibratory sensation. This process was repeated three times, for each of the feet. The patient’s global VPT was defined as the arithmetic mean of the measurements performed. A global VPT value greater than or equal to 6.4 micron/25 V was considered indicative of DPN [34,35,36]. If the patient did not perceive the vibratory sensation at the maximum of the scale (10.5 micron/32 V), the value was set as 32 V for statistical analysis. Patients with a difference of more than 7 V between the two feet or with an abnormal VPT on one foot and normal on the other were excluded from the study. Among the patients who accessed the centre, 201 met the inclusion criteria and were therefore enrolled in the study.

### 2.5. Statistical Analysis

Data were analysed with JMP^®^ Version Pro 17 software for Windows (SAS Institute Inc., Cary, NC, USA), JASP computer software (Version 0.18.3, JASP Team, University of Amsterdam, The Netherlands, 2024), and jamovi computer software (Version 2.5, The jamovi project, Sydney, Australia, 2024) with the latter two working in the R language (Version 4.3, R Core Team, Vienna, Austria, 2023). Continuous quantitative variables are expressed as means ± standard deviation. Missing data were handled using pairwise deletion to maximize the use of all available data. Statistical significance was assessed by one-way ANOVA or Student’s *t*-test for continuous data and by Pearson’s chi-square test for categorical variables. A *p* value < 0.05 was deemed statistically significant. Agreement between risk assessment obtained with AI software was tested against biothesiometer result (VPT), which was considered as reference for diagnosis of DPN. The percentage agreement between the two methods was calculated, as well as the Cohen’s κ value [37] and Gwet’s AC1 coefficient [38]. Benchmark references for Cohen’s κ were from Landis and Koch [39].

## 3. Results

Of the 201 patients enrolled, 107 (53.23%) were classified by AI software as having a low probability of developing DPN, 39 (19.40%) as having a moderate probability, 29 (14.43%) as having a high probability, and 26 (12.94%) as having a very high probability. In 63 of the total patients, biothesiometer measurements gave a VPT ≥ 25 V, indicative of DPN, while 138 patients presented a non-pathological VPT value (<25 V) (prevalence of abnormal VPT 31.34%; prevalence of normal VPT 68.66%). The mean VPT was significantly higher in the DPN group (29.7 ± 2.6 V) compared with the non-neuropathy group (mean VPT 14.9 ± 4.6 V, *p* < 0.0001).

The distribution of patients according to AI DPN risk classification and biothesiometer result is presented in Table 1. The overall distribution of patients with or without DPN, according to biothesiometer results, did not differ significantly across the four AI risk classifications (*p* = 0.0920, chi-square test). The VPT value did not differ as well, according to the AI risk classes within each group with or without DPN (*p* = 0.3093 and *p* = 0.9895, respectively, one-way ANOVA).

The clinical and biochemical characteristics of patients, divided by the presence or absence of neuropathy assessed with the biothesiometer, are illustrated in Table 2. Subjects with DPN showed a significantly longer mean disease duration than subjects without neuropathy (14.4 ± 8.9 y vs. 11.0 ± 8.4 y, respectively; *p* = 0.0110). A higher mean age was also observed in the DPN group (72.7 ± 6.7 y vs. 66.6 ± 9.9 y, respectively; *p* < 0.0001). Diastolic blood pressure was significantly lower in the DPN group compared with the non-neuropathy group (77.2 ± 10.5 mmHg vs. 80.5 ± 10.3 mmHg, respectively; *p* = 0.0356). No statistically significant differences were found in biochemical parameters.

In order to evaluate the concordance between the clinical outcome of DPN as a result of the biothesiometer test taken as reference method and the AI risk prediction of DPN, Cohen’s κ was calculated between the two parameters for each patient. The comparison was obtained, grouping the two classes “very high” and “high” risk (considered as positive—Yes code—for DPN occurrence) opposed to the other two classes, “medium” and “low” risk (considered as negative—No code—for DPN occurrence) according to the AI algorithm. Figure 1 presents the distribution of the concordance, with an overall agreement of 65% between biothesiometer result (positive—Yes code—if VPT ≥ 25 V) and AI risk attribution. Cohen’s κ was 0.162, a level significantly different from 0 (*p* = 0.021), but that must be considered as “slight” agreement if compared with the benchmarking proposed by Landis and Koch [39]. A “fair” degree of agreement was obtained with Gwet’s AC1 coefficient (0.405; 95%CI: 0.269–0.541), which is corrected in order to account for an expected disagreement rate as comparator [40].

## 4. Discussion

The prevalence of DPN in the subjects of this study (31.34%) is in line with the global epidemiology of the complication, the literature data being from 32% to 50% [5]. All the patients enrolled were asymptomatic for peripheral neuropathy, and approximately one-third of them showed an altered VPT despite the asymptomatic appearance, even with optimal glycaemic control. This is likely because DPN is more often characterized by negative symptoms (such as reduced pain, thermal, or vibration sensation), which patients frequently do not identify clearly on a subjective basis [5]. This finding suggests that virtually any patient with diabetes is at risk of developing DPN. The present study highlights the importance of regular instrumental screening to accurately assess the patient’s condition.

The main aim of the study was to assess the actual presence of neuropathy based on the risk assessment suggested by an algorithm of AI currently available as clinical support in Italian diabetology units. Overall, the classical indicators of concordance, namely, Cohen’s κ and Gwet’s AC1, indicate only a slight to fair agreement between the two rating methods. These indicators are very sensitive and able to correct for how often the raters may agree by chance [41], so it is not surprising that the level of concordance here estimated is quite low, possibly in part due to the intentionally wide margin of probability declared for risk estimation by the AI procedure. However, the degree of concordance between AI and the presence or absence of DPN, although not high (65%), was not due to chance.

Cohen’s κ is known to be influenced by the prevalence of the condition and by imbalanced marginal distributions. In settings with low to moderate DPN prevalence, κ can underestimate practical utility. Gwet’s AC1 adjusts for prevalence, and thus, provides a somewhat higher and possibly more stable estimate of the agreement. Taken together, the two measures frame the spectrum of “real-world” concordance. While perfect concordance with biothesiometry is neither expected nor required for a risk prediction tool, the observed performance suggests the AI module can serve as a triage instrument: patients flagged as “high” or “very high” risk can be prioritized for confirmatory biothesiometer testing and early intervention.

The likely low number of patients at high or very high risk may be one of the reasons why a strong concordance between the probability of onset and altered VPT was not found. A thorough exploration of these misclassifications will be the subject of future studies. We hypothesize that comorbid microvascular conditions may elevate the AI-predicted risk before sensory loss is clinically detectable and that the set of 25 clinical and biochemical parameters used by the algorithm [33] may have only moderate specificity. It should be noted that the calculation of the risk for DPN was based on traditional risk factors, which are not specific for neuropathy. In fact, comparing clinically defined DPN patients (diagnosed by biothesiometer) with non-neuropathic patients, the whole cluster of explored biochemical parameters (such as BMI, glycated haemoglobin, fasting blood glucose, total cholesterol, HDL cholesterol, LDL cholesterol, triglycerides), did not present any significant differences. In our study, the parameters (not used by the AI algorithm) that distinguished the two cohorts were the duration of disease and an older age, both significantly higher in patients with DPN. This result confirms what has been described in the literature, that the duration of disease and age are parameters significantly associated with the development of DPN, independently of other possible risk factors [12]. Our results align with a cross-sectional study utilizing magnetic resonance neurography in patients affected by T2D [42], which demonstrated that a longer duration of diabetes correlates with an increased burden of nerve lesions, indicating that such lesions may progressively accumulate and reach clinical significance when a critical number of nerve fascicles are involved.

In the literature, there are few references to the use of AI for DPN screening. Some authors have explored the classification of DPN using corneal confocal microscopy imaging data, showing high performance [25,26,27]. An AI-based study using high-resolution corneal confocal microscopy images to estimate the risk of diabetic neuropathy achieved promising results with an accuracy exceeding 80% [43]. However, the authors concluded that further multi-centre validation studies are needed. A recent study by Xiao et al. [44] proposed the use of toe photoplethysmography (PPG) signals, which reflect vascular dynamics and can predict both T2D and risk of peripheral neuropathy with accuracies rising from ~75–86% (Fisher, logistic regression) to ~93% (using artificial neural network learning method, ANN). However, the method is focused on vascular signals and not directly on nerve function, and PPG requires data acquisition for over 30 min, which can be difficult to achieve, especially with older subjects. Additionally, the study’s elderly cohort and single modality data may limit its generalizability to broader diabetic populations. Conversely, our approach combines the use of a well-trained AI algorithm on patients of different ages with fully independent external validation, here linked to a biothesiometer measurement that directly assesses large-fibre peripheral nerve function by testing vibration sense, therefore providing a quick neuropathy screening (loss of sensation). Moreover, the biothesiometer is an already validated method for neuropathy [17]; it is simple, immediate, easy read, and low-cost.

Regarding the present investigation, since the risk estimate is based on self-learning AI, it is possible that increasing the sample size could improve the algorithm performance, leading to a better concordance between AI risk classification and the occurrence of clinically diagnosed DPN. This study serves as a foundation for future research, particularly emphasizing the integration of additional parameters—such as disease duration and patient age—into the AI algorithm to enhance the accuracy of DPN risk estimation. It is also of note that the present study is among the first ones focused only on clinical and blood chemistry data, which are often used by physicians in clinical practice to evaluate the patient’s glyco-metabolic state and therefore are easily found in computerized medical records, in order to predict the risk of diabetes-related complications. Patients found to be at elevated risk of DPN may be prioritized for further evaluation using the biothesiometer test as a very simple instrumental screening, which requires limited personnel resource, also from non-medical healthcare professionals.

A limitation of this study is the use of biothesiometer as the sole reference method for DPN diagnosis. While practical and widely used, it may miss early or small-fibre neuropathy [36]. No additional confirmatory testing (e.g., nerve conduction studies) was performed. Future validation studies should incorporate more comprehensive diagnostic protocols to better assess the model’s performance across the full spectrum of DPN.

We acknowledge that excluding patients with symptoms of peripheral neuropathy may reduce the overall pre-test probability of DPN in the study sample and limit the generalizability to the broader clinical population. However, this was a deliberate choice to simulate a screening scenario, where the objective is to flag at-risk individuals before symptoms emerge. In such a context, high sensitivity and negative predictive value are critical, and the AI tool’s performance in these metrics supports its potential utility as an early risk stratification aid. The lack of significant differences in the distribution of AI risk groups between the two cohorts, distinguished based on biothesiometer VPT data using a 25 V cutoff, is also a consequence of selecting asymptomatic subjects. This approach was taken to allow for a more objective evaluation of the AI tool’s predictive performance, and it aligns with the tool’s intended clinical use: to support early risk stratification and prompt follow-up diagnostic assessments (e.g., via biothesiometry or neurologic evaluation), rather than to confirm DPN in symptomatic patients. Future studies in more heterogeneous populations, including symptomatic patients, will be important to further validate and refine the model.

While our results showed no significant differences in common biochemical markers between patients with and without neuropathy, this is consistent with previous evidence indicating that such parameters have limited discriminatory power for detecting DPN [45,46,47]. The AI algorithm applied in this study was developed and validated by Nicolucci et al. [23] based on a comprehensive set of demographic, clinical, and laboratory variables routinely collected in diabetes care. Although this approach ensures broad applicability in real-world settings, future iterations of the model could benefit from the inclusion of additional features more directly related to neuropathic risk. These might include assessments of small-fibre function, autonomic symptoms, long-term glycaemic variability, or foot thermoregulation patterns. However, integrating such parameters would require a new training and validation process using large, prospectively collected datasets. This represents a valuable direction for future refinement of the algorithm.

The use of AI in predicting DPN is a rapidly evolving area, with several AI models already being explored in the literature. Various machine learning techniques, such as convolutional neural networks (CNNs) and support vector machines (SVMs), have been applied to diverse datasets, including those based on photoplethysmography (PPG) signals, to predict diabetes-related complications [44]. However, these models may rely on features not routinely available in clinical settings, such as specialized imaging data or more comprehensive biochemical markers. Comparatively, our model focuses on using commonly collected clinical and biochemical data, which increases its potential for real-world applicability but may limit its predictive power in certain cases. Nonetheless, large-scale validation of these models, using diverse datasets from multiple centres, remains crucial to ensure their generalizability and robustness in clinical practice.

Practical implementation of AI tools in real-world diabetic clinics presents several challenges. Despite the growing potential of AI, its integration into existing healthcare systems faces barriers related to data privacy, interoperability with electronic health records, and clinician training [48]. Moreover, concerns about the cost and accessibility of AI-driven tools must be addressed to facilitate widespread adoption. Clinician acceptance is also key, as healthcare professionals must trust AI recommendations and view them as complementary to their clinical judgment rather than as replacements for human expertise.

From an ethical standpoint, the deployment of AI-based diagnostic tools raises important issues around bias in algorithmic predictions [49,50], especially when training datasets may not represent diverse populations adequately. Transparency in the validation of AI models and accountability for AI-driven decisions are essential for ensuring ethical integrity and maintaining stakeholder confidence in clinical settings [51]. Further, the use of AI in healthcare should ensure that clinicians remain central in the decision-making process, preserving the human element in diagnosis and treatment planning.

Diabetic neuropathy is an often-unrecognized complication of diabetes and, if it results in neuropathic ulcer, can significantly reduce the quality of life of patients [3,7]. The classical risk factors do not accurately predict the probability of developing DPN, and therefore, even a patient with optimal glycaemic control may still present with DPN. This complication of diabetes is, in fact, often diagnosed in asymptomatic patients through instrumental tests. The use of an optimized AI algorithm can help to estimate the risk of developing DPN by means of an integrated approach, which takes into account both the probability of onset calculated with the algorithm and the duration of the disease and age in order to better direct a more in-depth screening, including an instrumental approach by biothesiometer.

## Figures and Tables

**Figure 1 biomedicines-13-01075-f001:**
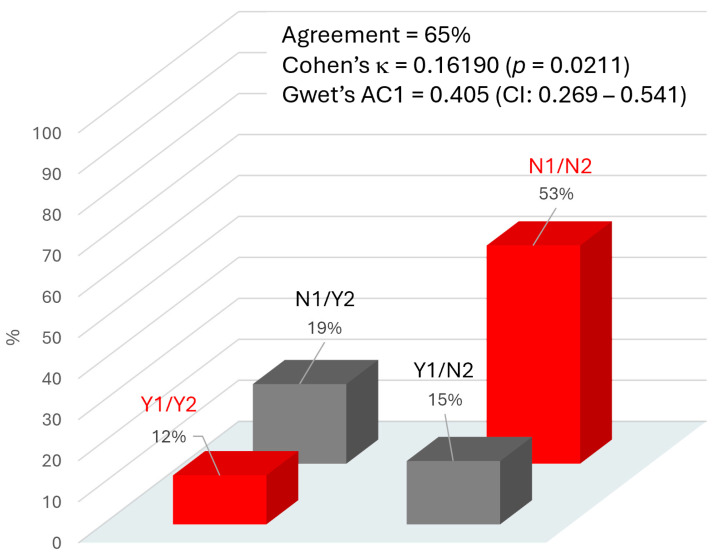
Agreement between the two methods used to evaluate DPN: AI software (Method 1) and biothesiometer testing (Method 2). Y1 (Yes-1) indicates a high probability of DPN, and N1 (No-1) indicates a low probability of DPN, according to the AI algorithm. Y2 (Yes-2) indicates the presence of DPN, and N2 (No-2) indicates the absence of DPN, as determined by the biothesiometer test (based on vibratory perception threshold, VPT, used as clinical reference). The percentage above each column in the graph represents the proportion of cases for each possible combination of agreement between the two methods. Red columns indicate agreement between the methods; grey columns indicate disagreement.

**Table 1 biomedicines-13-01075-t001:** Classification of patients in two cohorts, with or without DPN, according to biothesiometer results, and attribution of DPN risk by AI algorithm.

DPN Risk (AI Algorithm)	DPN Present (VPT ≥ 25 V)				DPN Absent (VPT < 25 V)				*p* *
	*n*	VPT (V)			*n*	VPT (V)			
Low	26 (41.27%)	29.23 ± 2.80	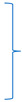	*p* = 0.3093 ^†^	81 (58.70%)	14.91 ± 4.82	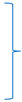	*p* = 0.9895 ^†^	*p* = 0.0920
Moderate	13 (20.63%)	30.23 ± 2.62			26 (18.84%)	14.88 ± 4.42			
High	13 (20.63%)	30.69 ± 1.84			16 (11.59%)	14.87 ± 4.29			
Very high	11 (17.46%)	29.18 ± 2.89			15 (10.87%)	14.47 ± 4.58			
Total	63 (100%)	29.73 ± 2.63			138 (100%)	14.86 ± 4.62			

* Significance in the distribution of patients across DPN risk groups based on the two biothesiometer cohorts, chi-square test. ^†^ Comparison of VPT values across DPN risk groups defined by AI algorithm, one-way ANOVA.

**Table 2 biomedicines-13-01075-t002:** Clinical and biochemical parameters of the patients divided by presence or absence of neuropathy assessed with the biothesiometer.

Parameter	DPN Present (*n* = 63)	DPN Absent (*n* = 138)	*p* *
Age, y	72.7 ± 6.7	66.6 ± 9.9	**<0.0001**
Sex, M/F	41/22	77/61	0.2150
Disease duration, y	14.4 ± 8.9	11.0 ± 8.4	**0.0110**
BMI, kg/m^2^	28.9 ± 4.8	28.3 ± 4.7	0.4369
Systolic blood pressure, mmHg	144.3 ± 18.5	139.6 ± 18.4	0.0910
Diastolic blood pressure, mmHg	77.2 ± 10.5	80.5 ± 10.3	**0.0356**
HbA1c, mmol/mol	52.6 ± 10.8	56.0 ± 14.1	0.0927
Fasting blood glucose, mg/dL	139.1 ± 38.0	137.6 ± 39.1	0.8023
Total cholesterol, mg/dL	161.6 ± 44.6	159.2 ± 39.0	0.7026
HDL cholesterol, mg/dL	50.0 ± 13.2	51.1 ± 14.6	0.6050
LDL cholesterol, mg/dL	91.5 ± 40.9	88.7 ± 29.9	0.5954
Triglycerides, mg/dL	114.7 ± 51.9	112.5 ± 58.3	0.7951
AST, IU/L	23.5 ± 10.0	24.9 ± 10.7	0.4560
ALT, IU/L	22.5 ± 10.9	25.1 ± 14.5	0.2709
Serum creatinine, mg/dL	0.97 ± 0.36	0.88 ± 0.30	0.0788
Microalbuminuria, mg/L	62.5 ± 136.1	53.2 ± 230.0	0.7811

* Significance was assessed with Student’s *t*-test for continuous variables and with Pearson’s chi-square test for categorical variables. Significant values are indicated in bold.

## Data Availability

The original contributions presented in this study are included in the article. Further inquiries can be directed to the corresponding author.

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
