# Peer review of "Artificial Intelligence Algorithm to Screen for Diabetic Neuropathy: A Pilot Study"

_biomedicines, 2025, doi:10.3390/biomedicines13051075_

Round 1
Reviewer 1 Report
Comments and Suggestions for Authors
The manuscript addresses an important clinical and scientific topic: the early detection of diabetic polyneuropathy (DPN) in type 2 diabetes (T2D) patients using an artificial intelligence (AI)-driven prediction algorithm in combination with a biothesiometer measurement as a reference. Given the global health burden of diabetic neuropathy and the emerging role of AI in clinical diagnostics, this study is timely and relevant. The concept of integrating routine clinical and biochemical data into AI-supported risk stratification tools could help streamline diabetic complication screening, optimize resource allocation, and promote early intervention. However, despite the potential clinical significance, the manuscript in its current form raises some concerns regarding methodology, interpretation of results, and depth of discussion. These issues need to be addressed to improve the clarity and robustness of the study before the work can be considered for publication.
Major Comments:
- While the use of MetaClinic AI software is interesting, the manuscript lacks a clear description of the algorithm's development and validation. Specifically:
- How was the AI model trained and on what dataset? Was this the same patient population or an independent dataset?
- How were thresholds for risk stratification (low, moderate, high, very high) established?
- Without this context, the reader cannot fully assess the reliability and reproducibility of the AI risk prediction. The related study (https://doi.org/10.1016/j.bspc.2021.103236) also used PPG and CNN for diabetes prediction. The authors should discuss relevant challenges and draw comparisons between the two studies to highlight strengths, limitations, and areas for improvement.
- The reported Cohen’s κ (0.162) and Gwet’s AC1 (0.405) suggest only slight to fair agreement between the AI risk classification and the biothesiometer-diagnosed DPN. While the authors mention this in the discussion, the manuscript lacks a clear explanation of the clinical relevance of this level of agreement. Could the authors elaborate on whether this degree of agreement is clinically acceptable for a screening tool? Additionally, was any analysis performed to identify why the AI algorithm misclassified a significant portion of the patients?
- The exclusion of patients with any symptoms of peripheral neuropathy appears to create a study population that is not representative of the broader clinical population where such an AI tool would be applied. This choice likely lowers the pre-test probability and might skew the performance of the AI model. The rationale for this design decision should be clarified.
- The biothesiometer is used here as the reference method, but this tool has known limitations in sensitivity and specificity, especially for early-stage neuropathy. Were any additional diagnostic tests (e.g., nerve conduction studies) performed to confirm the diagnosis? If not, this limitation should be clearly acknowledged.
- The study reports that common biochemical markers did not differ significantly between the neuropathy and non-neuropathy groups. However, this is unsurprising, as many of these variables have weak discriminatory power for neuropathy. Could the authors suggest what additional clinical features might improve the AI algorithm’s prediction if included? The discussion would benefit from addressing this point.
- While the authors highlight the need for early detection of DPN, the discussion could be enriched by including:
- Comparative perspectives with other AI models or machine learning approaches in the literature.
- Practical implementation challenges in real-world diabetic clinics.
- Ethical considerations when deploying AI-based diagnostic support tools.
Minor Comments:
- The manuscript is overall clear but would benefit from a professional language review to refine grammar and sentence structure for international readers. For example: Abstract line 14 — "patients with T2D those at risk of DPN" should be rephrased as "patients with T2D who are at risk of DPN."
- Table 1 and Table 2 present valuable data but could be improved, please include p-values for group comparisons in Table 1. Moreover, in Figure 1, the caption should explain all abbreviations fully and clearly.
- The ethical approval is appropriately documented, but the manuscript should mention whether data privacy and security protocols were applied in line with GDPR or equivalent standards, particularly since the study involves AI and personal medical data.
- The reference list is thorough and relevant but could benefit from a few more recent studies, particularly involving AI in diabetic neuropathy beyond corneal confocal microscopy (e.g., image-based, clinical decision-support systems, deep learning).
Author Response
Comments and Suggestions for Authors: The manuscript addresses an important clinical and scientific topic: the early detection of diabetic polyneuropathy (DPN) in type 2 diabetes (T2D) patients using an artificial intelligence (AI)-driven prediction algorithm in combination with a biothesiometer measurement as a reference. Given the global health burden of diabetic neuropathy and the emerging role of AI in clinical diagnostics, this study is timely and relevant. The concept of integrating routine clinical and biochemical data into AI-supported risk stratification tools could help streamline diabetic complication screening, optimize resource allocation, and promote early intervention. However, despite the potential clinical significance, the manuscript in its current form raises some concerns regarding methodology, interpretation of results, and depth of discussion. These issues need to be addressed to improve the clarity and robustness of the study before the work can be considered for publication.
Response: We thank the Reviewer for his/her thoughtful and constructive feedback and constructive critique. We sincerely appreciate his/her positive evaluation of the clinical and scientific relevance of our work. We have revised the manuscript to enhance methodological transparency, strengthen our interpretation, and deepen the discussion, and believe these changes have substantially improved the manuscript.
Major Comments:
Comment: While the use of MetaClinic AI software is interesting, the manuscript lacks a clear description of the algorithm's development and validation. Specifically:
How was the AI model trained and on what dataset? Was this the same patient population or an independent dataset?
Response: We thank the Reviewer for the opportunity to clarify the algorithm's development and validation details. The AI software for predicting diabetes complications has been originally described by Nicolucci and colleagues [Nicolucci et al., Diabetes Research and Clinical Practice 2022, 110013, DOI: 10.1016/j.diabres.2022.110013], who used XGBoost‑based models [Chen T, He T, Benesty M, Khotilovich V, Tang Y, Cho H, et al. Xgboost: extreme gradient boosting. R package version 04-2 2015;1(4):1–4] which were trained on the Smart Digital Clinic EMR data (Smart Digital Clinic, METEDA s.r.l) from 147 664 patients across 23 Italian diabetes centers over 15 years. For each of six complication groups (eye, cardiovascular, cerebrovascular, peripheral vascular, nephropathy, neuropathy), the model predicts the risk of onset within 2 years. To ensure true independence, models were externally validated on five additional different diabetes centers whose data were never part of the training set, thereby preserving generalizability and preventing data leakage. The sample size of the validation cohorts ranged 3,912 to 20,007 patients. Our study considered a totally independent population, consisting, as described in Methods, of consecutive patients recruited at the Diabetes Operating Unit of the ULSS 6 of Padua. In Methods section we added further information regarding the issue.
Comment: How were thresholds for risk stratification (low, moderate, high, very high) established?
Response: The AI Prediction Module of MetaClinic (METEDA S.r.l. MetaClinic User Guide: AI Prediction Module, Version 2.1. Milan, Italy: METEDA S.r.l.; 2022. Section 3, “Risk Stratification”) stratifies the risk of developing diabetic complications into four levels:
- Low risk: < 50th percentile
- Moderate risk: 50th–75th percentile
- High risk: 75th–90th percentile
- Very high risk: > 90th percentile
These categories are defined by applying three percentile cut‑offs (the 50th, 75th and 90th percentiles) to the calibrated risk probabilities in the short term (2 years) and medium term (3-5 years) for each complication, as described in the training and validation workflow. We added further information on the Methods section.
We trust this clarifies both the origin of the model (Nicolucci et al., 2022, DOI: 10.1016/j.diabres.2022.110013) and the data‑driven derivation of risk categories.
Comment: Without this context, the reader cannot fully assess the reliability and reproducibility of the AI risk prediction. The related study (https://doi.org/10.1016/j.bspc.2021.103236) also used PPG and CNN for diabetes prediction. The authors should discuss relevant challenges and draw comparisons between the two studies to highlight strengths, limitations, and areas for improvement.
Response: We thank the Reviewer for suggestion. We have added in the manuscript Discussion about the comparison of the two approaches: “ A recent study by Xiao et al. [44] proposed the use of toe photoplethysmography (PPG) signals which reflect vascular dynamics, can predict both T2D and risk of peripheral neuropathy with accuracies rising from ~75–86% (Fisher, logistic regression) to ~93% (using artificial neural network learning method, ANN); however, the method is focused on vascular signals, and not directly on nerve function and PPG requires data acquisition for over 30 min, which can be difficult to achieve, especially with older subjects. Additionally, the study’s elderly cohort and single‑modality data may limit generalizability to broader diabetic populations. Conversely, our approach combines the use of a well-trained AI algorithm on patients of different ages, with fully independent external validation, here linked to a biothesiometer measurement, which directly assesses large fiber peripheral nerve function by testing vibration sense, therefore providing a quick neuropathy screening (loss of sensation). Moreover, biothesiometer is an already validated method for neuropathy [17], it is simple, immediate, easy read out and of low cost.”
Comment: The reported Cohen’s κ (0.162) and Gwet’s AC1 (0.405) suggest only slight to fair agreement between the AI risk classification and the biothesiometer-diagnosed DPN. While the authors mention this in the discussion, the manuscript lacks a clear explanation of the clinical relevance of this level of agreement. Could the authors elaborate on whether this degree of agreement is clinically acceptable for a screening tool? Additionally, was any analysis performed to identify why the AI algorithm misclassified a significant portion of the patients?
Response: We thank the reviewer for highlighting the need to contextualize our agreement metrics and to explore the sources of discordance between the AI‐based risk classification and biothesiometer‐diagnosed DPN.
Cohen’s κ=0.162 indicates slight agreement and Gwet’s AC1=0.405 indicates fair agreement.
Cohen’s κ is known to be influenced by the prevalence of the condition and by imbalanced marginal distributions. In settings with low to moderate DPN prevalence, κ can underestimate practical utility. Gwet’s AC1 adjusts for prevalence and thus provides a somewhat higher and possibly more stable estimate of the agreement. Taken together, the two measures frame the spectrum of “real‐world” concordance. While perfect concordance with biothesiometry is neither expected nor required for a risk prediction tool, the observed performance suggests the AI module can serve as a triage instrument: patients flagged as “high” or “very high” risk can be prioritized for confirmatory biothesiometer testing and early intervention.
We did not conduct a dedicated analysis to determine why the AI algorithm misclassified a substantial proportion of patients, and at this stage such an investigation is challenging due to the small sample size. We hypothesize that comorbid microvascular conditions may elevate the AI‑predicted risk before sensory loss is clinically detectable, and that the set of 25 clinical and biochemical parameters used by the algorithm may have only moderate specificity. A thorough exploration of these misclassifications will be the subject of future studies.
We have incorporated these clarifications into the Discussion to better inform readers of both the utility and limitations of the AI risk prediction as a screening adjunct to biothesiometry.
Comment: The exclusion of patients with any symptoms of peripheral neuropathy appears to create a study population that is not representative of the broader clinical population where such an AI tool would be applied. This choice likely lowers the pre-test probability and might skew the performance of the AI model. The rationale for this design decision should be clarified.
Response: We thank the Reviewer for this important observation regarding our inclusion criteria. The decision to exclude patients with clinical symptoms of peripheral neuropathy was grounded by some considerations. First of all, our aim was to evaluate the AI tool as a screening instrument for asymptomatic individuals, in order to identify patients at high risk of DPN before overt symptoms appear. This aligns with the tool’s intended clinical application, which is to support risk stratification and guide earlier diagnostic follow-up (e.g., with biothesiometry or neurologic evaluation), rather than to confirm DPN in symptomatic patients. Including patients with established symptoms could introduce confirmation bias, as symptom-based clinical judgment might already influence decisions on neuropathy diagnosis. By selecting an asymptomatic cohort, we ensured a more objective evaluation of the tool’s predictive performance. We acknowledge the reviewer’s concern that excluding symptomatic patients may lower the overall pre-test probability of DPN, and this may indeed affect apparent performance metrics. However, we believe this is appropriate for a screening context, where the prevalence of disease is inherently lower than in diagnostic settings. We have added this rationale in Methods and Discussion sections.
Comment: The biothesiometer is used here as the reference method, but this tool has known limitations in sensitivity and specificity, especially for early-stage neuropathy. Were any additional diagnostic tests (e.g., nerve conduction studies) performed to confirm the diagnosis? If not, this limitation should be clearly acknowledged.
Response: We appreciate the Reviewer’s thoughtful comment. We fully acknowledge that while the digital biothesiometer is widely used in clinical practice, it has known limitations, particularly in detecting early or small-fiber neuropathy. No additional confirmatory diagnostic tests, such as nerve conduction studies or quantitative sensory testing were performed in this study. This was due to both logistical constraints and our intent to evaluate the AI risk prediction tool in a real-world, primary care–like setting, where such advanced diagnostics are typically unavailable. We have now acknowledged this limitation in the revised Discussion.
Comment: The study reports that common biochemical markers did not differ significantly between the neuropathy and non-neuropathy groups. However, this is unsurprising, as many of these variables have weak discriminatory power for neuropathy. Could the authors suggest what additional clinical features might improve the AI algorithm’s prediction if included? The discussion would benefit from addressing this point.
Response: We thank the Reviewer for this valuable suggestion. As noted, the limited discriminatory power of common biochemical markers for neuropathy is well recognized. The AI prediction algorithm used in our study was developed and validated in the multicenter study by Nicolucci et al. (2022), which incorporated a broad set of routinely available parameters, including demographic characteristics, diabetes duration, comorbidities, and a panel of standard laboratory tests. These features were selected for their availability across diverse clinical settings and their relevance to general diabetes management.
We agree that the inclusion of additional or more specialized clinical features, such as measures of small-fiber function, autonomic symptoms, foot temperature variability, or longitudinal glycemic variability, could potentially enhance the algorithm’s predictive accuracy for diabetic neuropathy, particularly in its early stages. However, such an extension would require access to larger datasets and a new training and validation cycle, ideally involving multimodal data. This represents a promising direction for future development of the tool. However, already in the present study, we found that the parameters age and duration of disease, which are not considered among AI parameters, might represent future implementation to the algorithm. We have included these perspectives in the revised Discussion to clarify the current model’s design and its potential for evolution.
Comment: While the authors highlight the need for early detection of DPN, the discussion could be enriched by including:
- Comparative perspectives with other AI models or machine learning approaches in the literature.
- Practical implementation challenges in real-world diabetic clinics.
- Ethical considerations when deploying AI-based diagnostic support tools.
Response: We thank the Reviewer for these thoughtful suggestions. We have revised the Discussion section to incorporate the suggested notes on AI models, practical implementation, and ethical considerations.
Minor Comments:
Comment: The manuscript is overall clear but would benefit from a professional language review to refine grammar and sentence structure for international readers. For example: Abstract line 14 — "patients with T2D those at risk of DPN" should be rephrased as "patients with T2D who are at risk of DPN."
Response: Thank you for your valuable suggestion. We have carefully reviewed the manuscript and revised the language to improve clarity and readability throughout.
Comment: Table 1 and Table 2 present valuable data but could be improved, please include p-values for group comparisons in Table 1. Moreover, in Figure 1, the caption should explain all abbreviations fully and clearly.
Response: According to suggestions, Table 1 was completed including p-values for group comparisons, to match with the description presented in Results section.
Table 2 was modified too. Clarity of the Legend to Figure 1 was improved.
Comment: The ethical approval is appropriately documented, but the manuscript should mention whether data privacy and security protocols were applied in line with GDPR or equivalent standards, particularly since the study involves AI and personal medical data.
Response: Thank you for this important observation. We confirm that the study was conducted in accordance with applicable data protection regulations, including the principles outlined in the General Data Protection Regulation (GDPR). The data used consisted exclusively of already available clinical records (e.g., anthropometric and biochemical parameters), along with the biothesiometer non-invasive measurement of peripheral nerve sensitivity performed at a single time point. No additional procedures or data collection beyond standard care were involved. All data were anonymized prior to analysis, and no identifiable personal information was used in the AI evaluation. This cross-sectional, observational study design ensured minimal risk to participants, and appropriate safeguards were implemented to maintain data security and confidentiality throughout. A specification was added in Methods section.
Comment: The reference list is thorough and relevant but could benefit from a few more recent studies, particularly involving AI in diabetic neuropathy beyond corneal confocal microscopy (e.g., image-based, clinical decision-support systems, deep learning).
Response: Additional references were quoted in the manuscript to highlight AI involvement in DPN.

Reviewer 2 Report
Comments and Suggestions for Authors
Dear authors,
Your study, "Artificial intelligence algorithm to screen for diabetic type 2 neuropathy: a pilot study," presents a valuable approach to applying AI for the prediction of human diseases, as demonstrated in the context of diabetic polyneuropathy (DPN). I strongly agree that this manuscript has the potential to contribute meaningfully to the field. Nevertheless, I would like to raise a few concerns.
Major Comments
1. From a predictive standpoint, it is crucial to clarify how early the model can anticipate the onset of the disease. According to your results, the system was able to classify patients into four groups based on the likelihood of developing DPN. However, within what timeframe does this classification apply? It is essential to clearly define the intended use of this AI tool: is it designed to enhance cost-effectiveness in diagnosing patients once the disease is clinically present, or to enable earlier detection prior to symptom onset?
2. While it is promising to demonstrate how an AI-based system can support clinical decision-making, the manuscript lacks clarity regarding the training of the algorithm. Please provide additional details about the model development and training process, including data sources, preprocessing steps, and validation methods.
3. Please include p-values for the comparisons in Table 1. Would it not be expected to observe significant differences between the groups? For instance, one might anticipate a higher proportion of 'very high' risk classifications in the DPN-positive group. These results should be more thoroughly discussed to assess the model’s discriminative power.
Author Response
Comments and Suggestions for Authors: Dear authors, Your study, "Artificial intelligence algorithm to screen for diabetic type 2 neuropathy: a pilot study," presents a valuable approach to applying AI for the prediction of human diseases, as demonstrated in the context of diabetic polyneuropathy (DPN). I strongly agree that this manuscript has the potential to contribute meaningfully to the field. Nevertheless, I would like to raise a few concerns.
Response: We sincerely thank the Reviewer for the encouraging feedback and for recognizing the potential impact of our work. We appreciate the thoughtful comments and have carefully addressed each of the concerns raised to further improve the clarity and quality of the manuscript. Our point-by-point responses to the specific issues are provided below.
Major Comments:
Comment: 1. From a predictive standpoint, it is crucial to clarify how early the model can anticipate the onset of the disease. According to your results, the system was able to classify patients into four groups based on the likelihood of developing DPN. However, within what timeframe does this classification apply? It is essential to clearly define the intended use of this AI tool: is it designed to enhance cost-effectiveness in diagnosing patients once the disease is clinically present, or to enable earlier detection prior to symptom onset?
Response: We thank the Reviewer for this important and insightful question. The AI tool used in our study is designed primarily for early risk stratification, with the goal of identifying individuals at increased risk of developing DPN even before clinical symptoms become evident. The classification into four probability-based groups reflects an estimated risk horizon of approximately 12 to 24 months, as indicated in the AI Prediction Module documentation (MetaClinic User Guide, METEDA S.r.l., Version 2.1, Section 3: “Risk Stratification”). This timeframe aligns with current evidence suggesting that early intervention within this window may improve outcomes and prevent progression to symptomatic neuropathy.
Furthermore, the approach is supported by data from Nicolucci et al. (Diabetes Research and Clinical Practice 2022, 110013, DOI: 10.1016/j.diabres.2022.110013), who emphasized the importance of predictive tools in the early identification of diabetic complications, particularly in asymptomatic individuals, to allow for targeted preventive strategies and cost-effective clinical management. Therefore, the AI model is intended not only to aid in the diagnosis of established DPN but more importantly to support proactive care by identifying patients at risk prior to symptom onset, thus contributing to improved clinical outcomes and healthcare efficiency.
We added information in the manuscript.
Comment: 2. While it is promising to demonstrate how an AI-based system can support clinical decision-making, the manuscript lacks clarity regarding the training of the algorithm. Please provide additional details about the model development and training process, including data sources, preprocessing steps, and validation methods.
Response: We thank the Reviewer for comment. The AI software for predicting diabetes complications has been originally described by Nicolucci and colleagues [DOI: 10.1016/j.diabres.2022.110013], who used XGBoost based models [Chen T, He T, Benesty M, Khotilovich V, Tang Y, Cho H, et al. Xgboost: extreme gradient boosting. R package version 04-2 2015;1(4):1–4] which were trained on the Smart Digital Clinic EMR data (Smart Digital Clinic, METEDA s.r.l) from 147 664 patients across 23 Italian diabetes centers over 15 years. For each of six complication groups (eye, cardiovascular, cerebrovascular, peripheral vascular, nephropathy, neuropathy), the model predicts the risk of onset within 2 years. To ensure true independence, models were externally validated on five additional different diabetes centers whose data were never part of the training set, thereby preserving generalizability and preventing data leakage. The sample size of the validation cohorts ranged 3,912 to 20,007 patients. Our study considered a totally independent population, consisting, as described in Methods, of consecutive patients recruited at the Diabetes Operating Unit of the ULSS 6 of Padua. In Methods section we added further information regarding the issue.
Comment: 3. Please include p-values for the comparisons in Table 1. Would it not be expected to observe significant differences between the groups? For instance, one might anticipate a higher proportion of 'very high' risk classifications in the DPN-positive group. These results should be more thoroughly discussed to assess the model’s discriminative power.
Response: We appreciate the Reviewer’s insightful comment. According to suggestions, Table 1 was completed including p-values for group comparisons, to match with the description presented in Results section.
The lack of significant differences in the distribution of AI risk groups between the two cohorts, distinguished based on biothesiometer VPT data using a 25 V cut-off, is a consequence of selecting asymptomatic subjects. This approach was taken to allow for a more objective evaluation of the tool’s predictive performance. Our primary aim was to assess the AI tool as a screening instrument for asymptomatic individuals, with the goal of identifying patients at high risk for DPN before overt symptoms emerge. This aligns with the tool’s intended clinical use: to support early risk stratification and prompt follow-up diagnostic assessments (e.g., via biothesiometry or neurologic evaluation), rather than to confirm DPN in symptomatic patients. We have added this rationale to the Methods and Discussion sections.

Round 2
Reviewer 1 Report
Comments and Suggestions for Authors
All comments have been carefully considered and addressed.
The recommendation is to publish the manuscript in its current form.
Reviewer 2 Report
Comments and Suggestions for Authors
Dear authors,
Your study, "Artificial intelligence algorithm to screen for diabetic type 2 neuropathy: a pilot study," presents a valuable approach to applying AI for the prediction of human diseases, as demonstrated in the context of diabetic polyneuropathy (DPN). I strongly agree that this manuscript has the potential to contribute meaningfully to the field. Thank you for responding to my previous questions.